# Pathogen-targeting glycovesicles as a therapy for salmonellosis

Haibo Mu[1,2], Hu Bai[1,2], Feifei Sun[1], Yinyin Liu[1], Chunbo Lu[1], Yuanhao Qiu[1], Peng Chen[1], Yu Yang[1], Lili Kong[1] & Jinyou Duan[1]

Antibiotic therapy is usually not recommended for salmonellosis, as it is associated with prolonged fecal carriage without reducing symptom duration or severity. Here we show that antibiotics encapsulated in hydrogen sulfide ($H_2S$)-responsive glycovesicles may be potentially useful for the treatment of salmonellosis. The antibiotics are released in the presence of *Salmonella*, which is known to produce $H_2S$. This approach prevents the quick absorption of antibiotics into the bloodstream, allows localized targeting of the pathogen in the gut, and alleviates disease symptoms in a mouse infection model. In addition, it reduces antibiotic-induced changes in the gut microbiota, and increases the abundance of potentially beneficial lactobacilli due to the release of prebiotic xylooligosaccharide analogs.

[1] Shaanxi Key Laboratory of Natural Products & Chemical Biology, College of Chemistry & Pharmacy, Northwest A&F University, 712100 Yangling, Shaanxi, China. [2] These authors contributed equally: Haibo Mu, Hu Bai. Correspondence and requests for materials should be addressed to J.D. (email: jduan@nwsuaf.edu.cn)

As our primary interface with the external environment, the gastrointestinal tract represents a vast mucosal surface area vulnerable to attack by enteropathogens[1]. Salmonellosis, caused by food-borne Salmonellae is one of the most frequently occurring intestinal diseases. Globally, there are ~153 million cases of gastroenteritis and 57,000 deaths each year[2].

The typical symptoms of salmonellosis, involving stomach cramps, nausea, and acute diarrhea, appear ~6–72 h after consumption of contaminated food or water[3]. This illness usually lasts 4–7 days, and most people recover without treatment. Current recommendations are to treat most patients with this illness with oral rehydration therapy but not with antimicrobial agents[4,5]. Antimicrobial therapy should be considered for patients who are severely ill (for example, those with severe diarrhea, high fever, or manifestations of extraintestinal infection) and for people at increased risk of invasive disease (infants, older adults, and the debilitated or immunosuppressed[6,7].

Antimicrobial therapy of salmonellosis is known to be associated with prolonged fecal carriage, without reducing symptom duration or severity, and may even increase the rates of long-term shedding of pathogens[8]. Systemically absorbed antibiotics such as quinolones and sulfonamides are readily taken up into the bloodstream if given orally, which makes them difficult to retain high concentrations at the site of enteric infections[9,10]. Poorly absorbed oral antibiotics such as aminoglycosides and β-lactam families allow localized enteric targeting of pathogens[11]. However, these antibiotics alter the balance of gut flora, leading to a possible overgrowth of opportunistic pathogenic bacteria[12].

To overcome the inherent defects of antimicrobial therapy in patients with salmonellosis, here we introduce a versatile vesicle-based system for delivering antibiotics to target the illness-causative agent, Salmonella enterica serovar Typhimurium selectively. This study might open up an avenue to develop pathogen-targeting antimicrobial glycovesicles to resolve enteric infections with a minimal risk of adverse outcomes.

## Results

### Design of pathogen-targeting antimicrobial glycovesicles.

Hydrogen sulfide ($H_2S$) production is considered a typical characteristic of Salmonella and an important marker for Salmonella isolation. Salmonella produce $H_2S$ from L-cysteine, by the activity of cysteine desulfhydrase[13] or produces $H_2S$ from thiosulfate by thiosulfate reductase[14]. Colonies that produce $H_2S$ are considered as the most clinical relevant and significant[13].

A $H_2S$-cleavable amphiphilic molecule (AM) is constructed by conjugation of xylooligosaccharide analogs with 1-dodecanethiol through a disulfide bond bridge (Fig. 1a and Supplementary Figs. 1–14). The spontaneous self-assembly of AMs forms spherical vesicles (Fig. 1b) with an average diameter of 100 nm (Fig. 1c), which elicit a clear Tyndall effect (Fig. 1b inset, right). The critical aggregation concentration of AM in water is calculated to be 0.0225 mg mL$^{-1}$ (Supplementary Fig. 16).

The spherical vesicles above collapse in the presence of 100 μM sodium sulfide ($Na_2S$) (Fig. 1b), accompanied with an increasing diameter (Fig. 1c) and a disappeared Tyndall effect (Fig. 1b inset, left). These data results imply that $Na_2S$ might result in the disassembly of AM vesicles, which is probably due to the decomposition of amphiphilic molecule caused by the reduction of disulfide bond.

The drug release behavior of AM vesicles was investigated using the fluoroquinolone antibiotic, ciprofloxacin hydrochloride (CIP) as a model drug, since fluoroquinolones are considered first-line treatment in salmonellosis[15]. From UV/Vis absorption spectra, the CIP loading efficiency is 9.2% (w/w), indicating a good drug-loading capability of AM vesicles. There are only about 2%, 4%, and 6% CIP released, respectively, from CIP-loaded AM vesicles (AM-CIP) when incubated 4 h in PBS, simulated gastric fluid (SGF, pH 2) and simulated intestinal fluid (SIF, pH 7) (Fig. 1d). Since the mean gastric emptying time of adults is less than 4 h[16], this finding indicates AM-CIP is fairly stable in gastrointestinal environment when given orally. In contrast, CIP is released from AM-CIP quickly and the released CIP reaches to ~50% at 4 h later in the presence of $Na_2S$ (100 μM), which is consistent with the previous observation that $Na_2S$ induces the disassembly of AM vesicles. Indeed, $Na_2S$ induces CIP release from AM-CIP in a dose-dependent manner (Supplementary Fig. 17).

To see whether pathogens can decompose AM vesicles, the $H_2S$-producing pathogen, S. typhimurium is co-cultured with AM-CIP. AM-CIP at low concentrations (<3.28 μg mL$^{-1}$) has a weaker bactericidal capacity than the same amount of CIP does, while their activities are comparable at high concentrations (Fig. 1e). This data impliy that S. typhimurium can produce sufficient $H_2S$ to decompose AM-CIP, which enables the direct contact between the released CIP and pathogens. The $H_2S$ concentration produced by S. typhimurium ($10^6$ CFU mL$^{-1}$) is determined to be 125 μM, according to the colorimetric method[17].

An $H_2S$-responsive fluorescence probe, Fluo (Supplementary Fig. 15) is also employed to evaluate whether the disassembly of AM vesicles is dependent on specific bacteria. There are far stronger fluorescence intensity in the Fluo-loaded AM vesicles when incubated with S. typhimurium (producing $H_2S$) than that of S. paratyphi A (not producing $H_2S$) (Fig. 1f), an observation that AM vesicles can only be decomposed by $H_2S$-producing organisms such as S. typhimurium selectively. Although other $H_2S$-producing organisms such as C. freudii also induces mild fluorescence at the same cell density as S. typhimurium (Fig. 1f), the high abundance of S. typhimurium in infected intestines will facilitate the disassembly of AM vesicles by this pathogenic bacteria.

### Pathogen-targeting glycovesicles treat salmonellosis efficiently.

An acute intestinal infection mouse model is developed using Salmonella enterica serovar Typhimurium (Fig. 2a). Salmonella infections result in a constant decline of body weight (Fig. 2b). CIP treatment can partially prevent weight loss and increase food uptake, while AM-CIP has a more pronounced effect than CIP does (Fig. 2b, c). CIP therapy leads to a decrease of Salmonella counts in the gastrointestinal tract (Fig. 2d, e), but it doesn't affect pathogen burden at the extraintestinal sites such as liver and spleen at the tested doses (Fig. 2f, g). In contrast, AM-CIP can efficiently reduce S. typhimurium infections in the gastrointestinal tract and thereby alleviate its dissemination into liver and spleen (Fig. 2d–g). In line with the above observation, severe tissue destruction and inflammatory infiltrates are observed in enteric mucosa and submucosa of small intestines and colons from untreated S. typhimurium infected mice (Fig. 2h). These epithelial damage and inflammation are greatly alleviated after CIP-AM treatment, which is superior to that of CIP.

To find out drug release behavior in the intestine, infected or normal mice are orally administrated with Fluo-loaded AM vesicles. The ileum from infected mice, but not from normal ones elicits the strongest fluorescence intensity in a time-dependent manner (Fig. 2i), implying that 1) S. typhimurium triggers the release of the probe from AM vesicles and 2) there is higher relative abundance of S. typhimurium in ileum. Further, infected mice are administrated with fluorescence (5-DTAF)-labelled S. typhimurium or Fluo-loaded AM vesicles. The fluorescence-labelled bacteria mainly retain in ileum (Fig. 2j), even after 12 h post oral administration, an observation that the ileum is one primary site

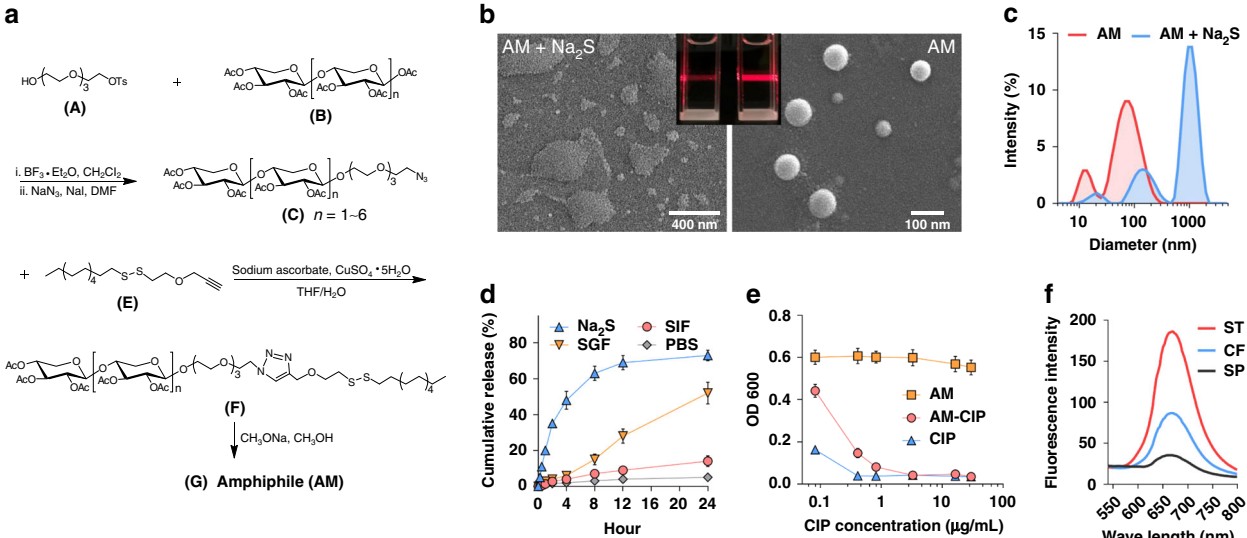

**Fig. 1** Synthesis and characterization of AM vesicles. **a** Synthetic route of amphiphile (AM). **b** Scanning electron microscopic (SEM) images of AM vesicles with (left) or without (right) Na₂S (100 µM). Scale bars represent 400 nm (left) and 100 nm (right), respectively. Inset: Tyndall effect of AM vesicles with or without Na₂S. **c** Hydrodynamic size distribution from DLS analysis in the absence or presence of Na₂S (100 µM). **d** Cumulative release profiles of AM-CIP in PBS, PBS with Na₂S (100 µM), simulated gastric fluid (SGF), or simulated intestinal fluid (SIF). Each is performed in triplicate. **e** In vitro bactericidal activity of free CIP (CIP), CIP-loaded AM vesicles (AM-CIP), and blank AM vesicles against *S. typhimurium* bacteria. Each is performed in triplicate. **f** Fluorescence intensity of Fluo-loaded AM vesicles (containing 10 µM Fluo) incubated with *S. typhimurium* (ST), *C. freudii* (CF), or *S. paratyphi* A (SP) (10⁶ CFU mL⁻¹) for 12 h. Data are means ± SD, representative of three technical repeats. Source data are provided as a Source Data file

of *Salmonella* infection. Consistently, there is strongest fluorescence intensity in ileum after administration of Fluo-loaded AM vesicles (Fig. 2j). Taken together, these findings indicate that AM vesicles are disassembled on-demand at the site of infection.

Unlike poorly absorbed oral antibiotics, absorbable antibiotics such as the fluoroquinolone family cannot retain therapeutic concentrations at the site of enteric infection[10,11]. As expected, CIP is readily absorbed and taken into the bloodstream in 30 min after oral administration. Differently, there are a little amount of CIP detected in the blood and most CIP remains in the intestine after 4 h if orally given with AM-CIP (Fig. 2k, l). This data indicate that AM vesicles can prevent the quick absorption of systematically available antibiotics into the bloodstream.

**Glycovesicles reduce antibiotic-induced changes in gut microbiota.** One common side effect of antibiotic treatment for enteric infections is collateral damage to the gut microbiota composition, resulting in a possible overgrowth of opportunistic pathogenic bacteria. Now it is believed that a repeated exposure to therapeutic doses of antimicrobials can even lead to long-lasting disruption of the gut flora and this side effect is not restricted to orally applied antibiotics[18,19].

To assess the effect of AM-CIP on gut microbial communities, we perform a pyrosequencing-based analysis of bacterial 16 S rRNA in caecal feces from *S. typhimurium* infected mice treated with water, AM, CIP, or AM-CIP (Fig. 3a). Similar abundance of the *Bacteroidaceae* family is observed in all groups (Fig. 3b). One most striking different is that CIP treatment induces an overgrowth of the *Lachnospiraceae* family (Fig. 3c). Several members from this family are considered opportunistic pathogens, which have been identified in inflamed samples such as diabetic wounds[20], irritable bowel syndrome[21,22], subgingival crevice[23], and cystic fibrosis[24]. The other difference is the variance of abundance of the *Lactobacillaceae* family for each treatment. Both AM-CIP and AM treatment, but not CIP greatly increase abundance of the *Lactobacillaceae* family (Fig. 3d), which is probably due to the prebiotic xylooligosaccharide

analogs released after collapse of AM vesicles. This is encouraging since several strains from the *Lactobacillaceae* family are shown to be highly antagonistic to *Salmonella* pathogens and protect against *Salmonella* infections in the gastrointestinal tract[25–27]. It is reasonable that AM treatment does not increase *Lactobacillaceae* as much as AM-CIP treatment does, due to the suppressive effect of abundant *Salmonella* pathogens on *Lactobacillaceae* in AM-treated groups. To see whether AM-CIP shaped gut microbiota is beneficial to the host, we transfer the gut microbiota from AM-CIP or CIP-treated *Salmonella*-infected donor mice to infected recipient mice. The recipient mice received with AM-CIP-shaped gut microbiota have lower pathogen burden and tissue inflammation than that in mice received with CIP-shaped one (Supplementary Fig. 18).

To further define whether this strategy can minimize antibiotic-induced collateral damage to gut homeostasis, one poorly absorbed antibiotic, neomycin (NEO) is encapsulated to yield neomycin-loaded AM vesicles (AM-NEO). Although total NEO content in the intestine is quite similar whatever NEO or AM-NEO is given (Supplementary Fig. 19), as expected, NEO treatment significantly increased the relative abundance of *Bacteroidales* and *Clostridiales* and decreased content of *Lactobacillales* and *Campylobacterales*. In contrast, AM-NEO treatment has not those effects (Fig. 4a, b). The microbiota structural changes were then analyzed using unsupervised multivariate statistical methods including UniFrac distance-based principal coordinate analysis (PCoA) and nonmetric multidimensional scaling (NMDS) (Fig. 4c, d). NEO treatment presented a centralized clustering of microbiota composition, indicating NEO reduces microbiota diversity. On the contrary, AM-NEO treatment does not alter gut microbiota composition as NEO treatment does.

**Discussion**

Although antibiotics are a life-saving tool for the treatment of bacterial infections, antimicrobial therapy is not routinely recommended for patients with enteric infections[8]. Antibiotic

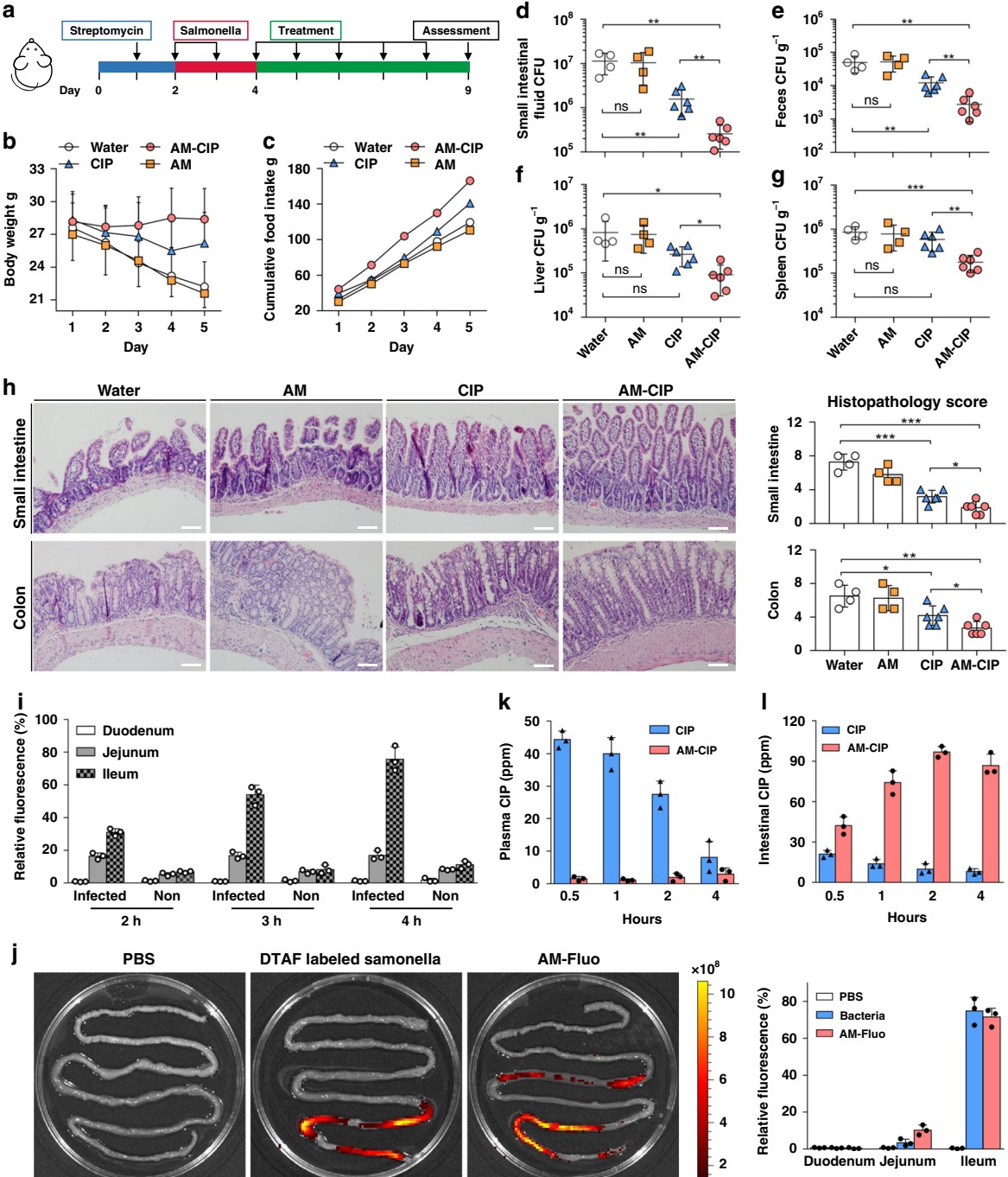

**Fig. 2** Therapeutic efficacy in acute intestinal infection model. **a** The study protocol including streptomycin pretreatment and *S. typhimurium* inoculation on mice, followed by the treatments [AM (100 mg kg⁻¹), CIP (10 mg kg⁻¹), or AM-CIP (110 mg kg⁻¹) daily]. **b** Body weight. $n = 4$–6 mice per group. **c** Cumulative food intake. $n = 4$–6 mice per group. Quantification of bacterial burden in the small intestine (**d**), feces (**e**), liver (**f**), and spleen (**g**) of infected mice with different treatments. **h** H&E stained sections of intestine tissues from infected mice after with individual treatments. Scale bar, 100 μm. Histogram represents combined histopathology score ($n = 4$–6 mice per group). **i** Fluorescent intensity of small intestines from normal or infected mice 2 h, 3 h, and 4 h after administration of Fluo-loaded AM vesicles ($n = 3$ mice per group). **j** Fluorescence images of small intestines from infected mice 12 h after administration of PBS, 5-DTAF-labeled *S. typhimurium* or Fluo-loaded AM vesicles ($n = 3$ mice per group). **k, l** Mean plasma or intestinal concentration of CIP following a single oral administration of AM-CIP (110 mg kg⁻¹) or CIP (10 mg kg⁻¹) in normal mice ($n = 3$ mice per group). Data are shown as mean ± SD, and each dot represents one animal. A single asterisk indicate p-values of <0.05, double asterisks indicate *p*-values of <0.01, triple asterisks indicate *p*-values of <0.001, *ns* no statistical significance (two-tailed Student's *t*-test). Source data are provided as a Source Data file

use will induce important collateral damage to host-associated microbial communities[28,29]. This puzzle prompts us to develop a strategy to eliminate enteric pathogens with minimized negative influence on gut microbiota during antimicrobial therapy.

Disulfide bonds have been widely used to develop reduction-responsive drug-delivery systems for cancer therapy, since it can be rapidly degraded and selectively release cargoes in tumor tissues, which contain higher glutathione levels than that in normal tissues[30]. To our interest, the disulfide bond can also be easily

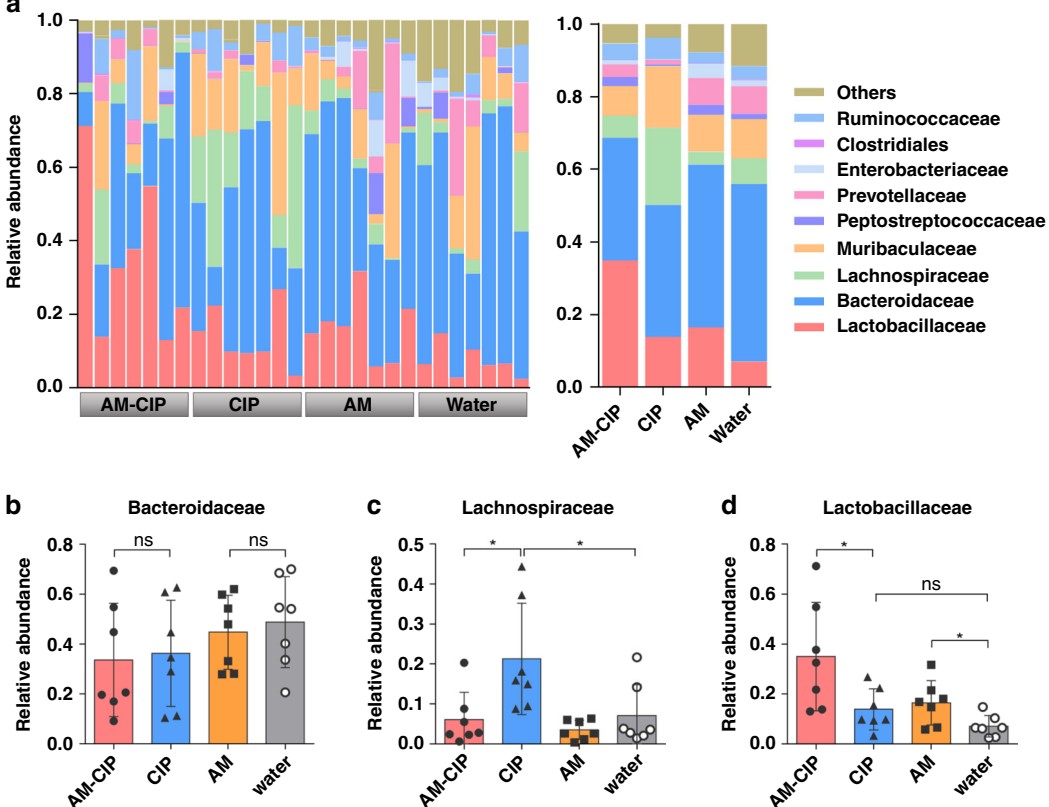

**Fig. 3** Gut microbial analysis in infected mice after treatment. **a** Left: bacterial taxonomic profiling at family level of the gut microbiota in feces from infected mice post therapy. Right: average composition of the family level in four groups. Relative abundance of *Bacteroidaceae* (**b**), *Lachnospiraceae* (**c**), and *Lactobacillaceae* (**d**) obtained in fecal microbiota from the LefSe results. Data are shown as mean ± SD, $n = 7$ mice per group and each dot represents data from an individual animal. A single asterisk indicate $p$-values of <0.05, ns, no statistical significance (two-tailed Student's $t$-test). Source data are provided as a Source Data file

broken down by $H_2S$, a Janus-faced molecule which can be produced selectively by several sulfate-reducing bacteria such as *Streptococcus*, *Fusobacterium*, *Salmonella*, *Enterobacter*, and *Helicobacter* in the gastrointestinal tract[31–33].

In this study, xylo-oligosaccharides are conjugated with a long-chain fatty chain by a disulfide bond bridge to form an amphiphilic molecule, whose self-assembly can generate an $H_2S$-responsive antibiotic delivery system. In vivo toxicity evaluation indicates that this drug-delivery system is biocompatible and safe during oral administration in mouse models (Supplementary Fig. 20).

By using a well-established acute *Salmonella* infection model, we find that this system can prevent the quick absorption of ciprofloxacin in the small intestine and release the antibiotic at *Salmonella*-rich infectious sites. The releasing antibiotic is uptaken by *Salmonella* pathogens nearby and thus dramatically decrease bacterial burden. After cleavage of the disulfide bond, the generated xylooligosaccharide analogs can promote the growth of the beneficial microbiota, *Lactobacillaceae* in the gastrointestinal tract (Fig. 5). It is known that oligosaccharide prebiotics such as xylooligosaccharide could only be utilized by a limited number of beneficial bacteria including lactobacilli and selectively proliferate these organisms[34].

Most commercially available antibiotics including ciprofloxacin are easily absorbed and are rapidly taken up into the bloodstream, which make them unfavorable for localized enteric targeting of pathogens in the gastrointestinal tract[9,10,35]. This drug-delivery system retrieves absorbable antibiotics above as a drug reservoir to treat enteric infections caused by $H_2S$-producing pathogens, which might be extremely important in case of experiencing highly virulent antibiotic-resistant infections in the gastrointestinal tract.

This drug-delivery system is further employed to load one poorly absorbed antibiotic to find out whether it can ameliorate antibiotic-induced collateral damage to the gut microbiota composition. It is intriguing that this delivery system does not disturb the microbiota greatly as the poorly absorbed antibiotic alone does (Fig. 4).

In summary, we develop an $H_2S$-responsive antibiotic delivery system for selective targeting of *Salmonella*, the salmonellosis-causative pathogen. This system includes the following features: (1) prevent quick absorption of oral antibiotics into the bloodstream; (2) allow antibiotics to target enteric pathogens locally and thereby alleviates disease symptoms; (3) ameliorate antibiotic-induced damage to gut microbiota greatly; and (4) increase beneficial microbiota *Lactobacillaceae* by the releasable prebiotic xylooligosaccharide analogs. This study might open up an avenue to develop pathogen-targeting antimicrobial glycovesicles to resolve enteric infections.

## Methods

**Chemical reagents**. Tetrathylene glycol (99%), Tosyl chloride (99%), Sodium acetate (≥99%), Acetic anhydride (≥98.5%), Sodium azide (99.5%), Pyridine (99%) were obtained from Aladdin Industrial Inc. (Shanghai, China). Xylo-oligosaccharides (95%) was purchased from Shanghai Yuanye Biological Technology Co., Ltd. (Shanghai, China). Dodecane-1-thiol (≥95%), 2-mercaptoethanol (99%) were bought from TCI (Shanghai, China). Propargyl bromide (99%) was obtained from Energy Chemical Reagent Co. All other reagents were of analytical grade and used as received. NMR spectra were recorded on a Bruker 500 MHz Spectrometer with working frequencies of 500 MHz for $^1$H and 125 MHz for $^{13}$C,

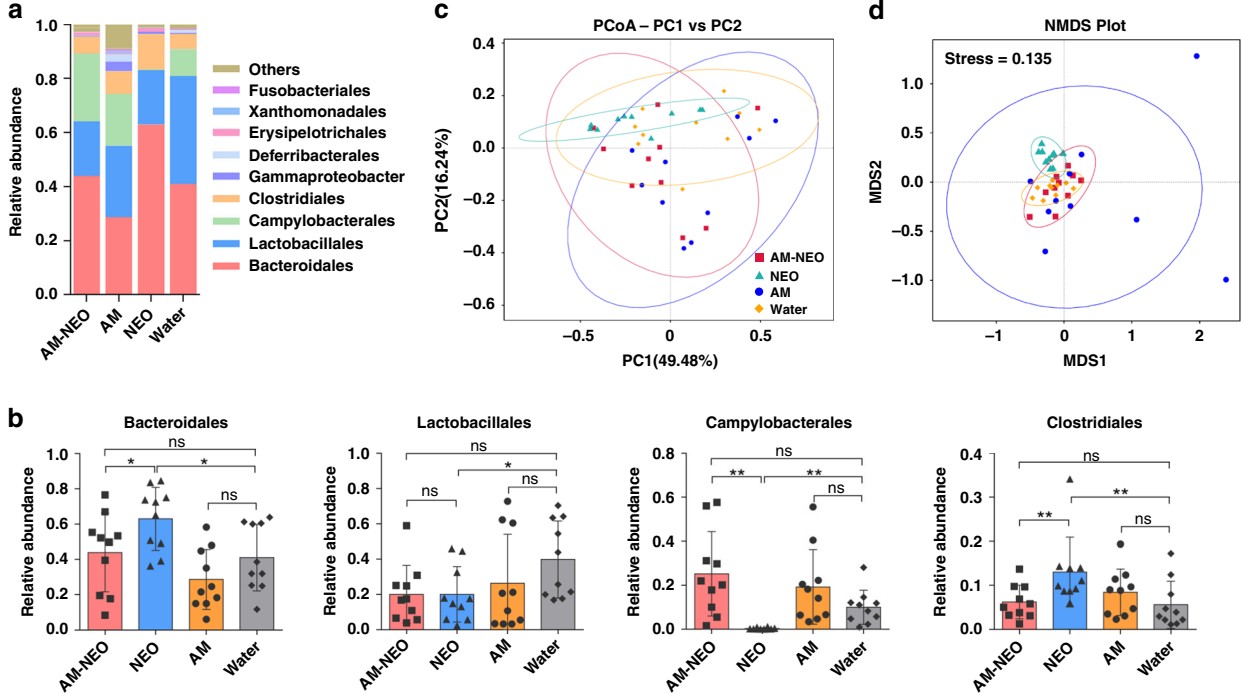

**Fig. 4** AM vesicle cloak alleviates neomycin induced damages to the gut microbiota in normal mice. **a** Average bacterial taxonomic profiling at order level of the gut microbiota in four groups. **b** Relative abundance of the bacterial order obtained in fecal microbiota from the LefSe results. Data are shown as mean ± SD, $n = 10$ mice per group and each dot represents data from an individual animal. A single asterisk indicate $p$-values of <0.05, double asterisks indicate $p$-values of <0.01, ns, no statistical significance (two-tailed Student's $t$-test). Source data are provided as a Source Data file. **c** Nonmetric multidimensional scaling score plot based on Bray-Curtis. **d** UniFrac-based PCoA score plot based on weights

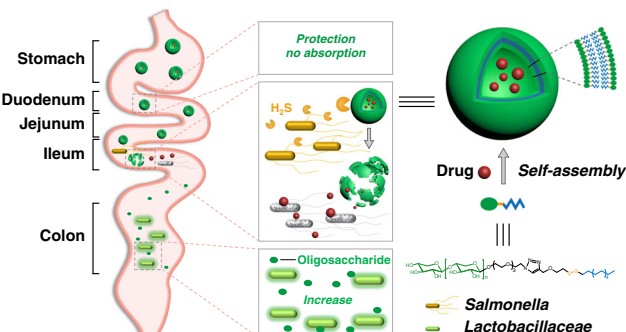

**Fig. 5** A proposed model for how AM vesicles resolve salmonellosis efficiently. The envisioned paradigm of AM vesicles to protect ciprofloxacin (CIP) from absorption, eliminate *Salmonella* bacteria locally and promote *Lactobacillaceae* growth

respectively, in CDCl₃ or CD₃OD. The residual signals from CDCl₃ (¹H: δ 7.26 ppm; ¹³C: δ 77.00 ppm) or CD₃OD (¹H: δ 3.31 ppm; ¹³C: δ 49 ppm) were used as internal standards. HRMS were acquired using a high resolution mass spectrometer (New ultrafleXtreme; Bruker Daltonik, Bremen, Germany). Ions were generated with a pulsed 337 nm nitrogen laser and accelerated to 25 kV. All spectra were obtained in the reflectron mode with delayed extraction of 200 ns. For sample preparation, 0.5 μL of 2,5-dihydroxybenzoic acid (DHB) (10 mg mL⁻¹) in 30% ethanol was spotted onto a target plate (MTP 384 target plate ground steel, Bruker Daltonik). After dried, an aliquot (0.5 μL) of the sample solution was spotted onto the DHB crystal and dried. All HRMS spectra were obtained from Na⁺ adduct ions.

**Syntheses of AM**. The synthetic details and characterization of compound A–G can be found in Supplementary Note 1 and Supplementary Figs. 1–14. The synthesis and characterization of Fluo were shown in Supplementary Fig. 15.

**The preparation, characterization, and stability of the vesicles**. Three milligram of amphiphile (AM) were dissolved in 1 mL methanol, and then the organic

solvent was evaporated to form a dried lipid film. The lipid film was rehydrated with 3 mL of deionized water, or 1 mg rhodamine B (RhB), or 3 mg ciprofloxacin (CIP), followed by vortexing for 1 min and sonicating for 30 min to produce AM-CIP vesicles, purified by dialysis (molecular weight cutoff 3500 Da) in deionized water. The unencapsulated CIP in the dialysate was quantified by UV-Vis at 277 nm. For preparation of probe-loaded vesicles, 0.2 mg of Fluo was dissolved in methanol with 3 mg of AM before evaporation.

**Release behaviors**. The kinetics of ciprofloxacin release was studied from the prepared AM-CIP. The fresh prepared AM-CIP solution (2 mg mL⁻¹) was initially incubated with or without Na₂S in PBS (SIP or SGF) at 37 °C. At predetermined time points, released CIP was separated by filtration using 10 kDa MWCO Amicon centrifugal filters (EMD Millipore, Billerica, CA, USA). CIP concentration was determined by measuring the absorbance at 277 nm.

**Bacteria strains**. *Salmonella enterica serovar typhimurium* SL1344, *Salmonella paratyphi* A ATCC9150, and *Citrobacter freundii* ATCC43864 were purchased from BeNa Culture Collection (Beijing, China). The bacteria strains were routinely cultured in Luria-Bertani (LB) broth at 37 °C with moderate reciprocal shaking overnight for future use.

**In vitro bactericidal activity**. Bacteria samples (10⁶ CFU mL⁻¹) were mixed well with LB including different concentrations of CIP (0.082–29.820 μg mL⁻¹), equivalent AM-CIP or AM. After incubation at 37 °C for 24 h, the OD₆₀₀ was monitored.

**Mouse strains and husbandry**. Kunming mice were obtained from Xi'an Jiaotong University. Female mice (5 weeks old) were housed in cages containing sawdust bedding in holding rooms with a temperature 25 °C and a relative humidity of about 40%. Deionized water and food were provided ad libitum. All animal procedures complied with all relevant ethical regulations and were approved by the Northwest A&F University Animal Care Committee (NWAFU-314020038). Mice were given 2 week to acclimate before experimentation.

**Mouse infections**. As described[8], water and food were withdrawn 4 h before per os treatment with 20 mg of streptomycin. Afterward, animals were supplied with water and food ad libitum. At 20 h after streptomycin treatment, water and food were withdrawn again for 4 h before the mice were infected via oral gavage with 10⁹ CFU of *S. typhimurium* in 100 μL PBS. Thereafter, drinking water food ad libitum

was offered immediately. One day later, mice received oral gavage with water, AM (100 mg kg$^{-1}$), CIP (10 mg kg$^{-1}$), or AM-CIP (110 mg kg$^{-1}$) once every day for 5 days.

**S. typhimurium burden in tissues**. After collection of fresh fecal pellets, animals were euthanized by cervical dislocation. The intestine, spleen, and liver were collected in sterile PBS. Homogenates were then serially diluted and plated onto LB agar to enumerate colony-forming units (CFU).

**Gut microbiota profiling**. Fresh fecal pellets were collected, stored in liquid nitrogen, transported to Novogene (Beijing, China), packed with dry ice, and then immediately stored in a −80 °C refrigerator before extraction of total DNA. The 16 S rRNA gene comprising V3–V4 regions was amplified using common primer pair and the microbial diversity analysis was performed as described[36]. Briefly, the raw sequences were first quality-controlled using QIIME with default parameters, then demultiplexed and clustered into species-level (97% similarity) operational taxonomic units (OTUs). OUT generation is based on GreenGene's database and the reference-based method with SortMeRNA. Strain composition analysis, alpha diversity analysis and beta diversity analysis were performed using QIIME. Discriminative taxa were determined using LEfSe (LDA Effect Size).

**Fecal transplantation**. Fecal transplant was performed as a reported protocol[37]. Briefly, 8-week-old female donor mice ($n = 5$ mice per group) were infected and treated with CIP or AM-CIP for 5 days as previous. Then stools were collected daily for the subsequent 6 days under a laminar flow hood in sterile conditions. Stools from donor mice of each treated group were pooled and 100 mg was resuspended in 1 mL of sterile PBS with vigorous mixing, and then subjected to centrifugation at 600 g for 5 min. The supernatant was collected and used as transplant material. Fresh transplant material was prepared on the same day of transplantation within 10 min before oral gavage to prevent changes in bacterial composition. Eight-week-old female recipient mice ($n = 9$–10) were infected beforehand and inoculated daily with fresh transplant material (200 µL for each mouse) by oral gavage for 6 days, before being killed for subsequent analysis.

**Blood and intestine drug concentration**. As previously described[38], 7-week-old uninfected Kunming female mice were randomly divided into two groups ($n = 12$). All experimental animals received intragastrically a single dose (300 µL) of CIP or AM-CIP. After treatments, three mice per group were humanly sacrificed at each time point post-administration 0.5, 1, 2, and 4 h. Blood samples (800 µL) were collected into heparinized tubes. The samples were centrifuged for 15 min (4 °C, 2000 rpm), and the plasma was separated and transferred into polypropylene tubes. The small intestines were also collected and flushed with PBS. The ciprofloxacin concentrations in blood and intestine were then determined by ELISA kit (ZIKER Biological Technology, Shenzhen, China).

**In vivo intestinal tract site-specific response**. Seven-week-old infected Kunming female mice were gavaged with 300 µL of AM-Fluo ($n = 3$ mice per group). Intestinal tracts including duodenum, jejunum, and ileum from each mouse were collected at indicated time points (2, 3, 4 h) after administration. The tissues were rinsed inside with DMSO, and the consequent DMSO was subjected to fluorescence determination using a fluorescence spectrophotometer, Ex/Em = 520/670 nm. For in vivo images, *S. typhimurium* were labeled using the dye 5-(4–6-dichlorotriazinyl) aminofluorescein (5-DTAF)[39]. Infected mice ($n = 3$ per group) were orally gavaged with 200 µL suspension of 5-DTAF-labeled bacteria ($10^9$ CFU) containing 0.2 M NaHCO₃, or administered orally with 200 µL suspension of AM-Fluo, PBS as a control. At 12 h after administration, the intestinal tracts were dissected, rinsed with PBS, and then imaged using an in vivo imaging system (IVIS, Lumina XRMS III, PerkinElmer). Fluorescence intensity of images were calculated using Living Image software from Caliper Life Sciences.

**In vivo toxicity study**. To evaluate the acute toxicity of the AM-CIP in vivo, 7-week-old uninfected Kunming female mice were orally administered with AM-CIP once daily for 5 consecutive days. Mice administered with deionized water were tested in parallel as a negative control. During the experimental period, the mouse body weight and food intake were monitored by weighing the mice daily. On day 6, mice were killed and sections of the small and large intestine tissues were processed for histological examination. The small and large intestines were cut to small sections as duodenum, jejunum, ileum, and colon and rinsed inside with PBS to remove internal residues. The longitudinal tissue sections were fixed in neutral-buffered 10% formalin for 15 h, transferred into 70% ethanol, and embedded in paraffin. The tissue sections were cut with 5 µm thickness and stained with H&E assay.

**Histopathology scoring**. Histopathology scoring was performed on H&E stained small intestine (ileum) and colon tissue sections according to the criteria below: (a) Integrity of the intestinal epithelium: intact and no pathological changes (0), mild (1), moderate (2), and severe destruction (3); (b) Mucosal inflammation: no inflammatory infiltrates (0), rare (1), moderate (2), and massive invasion of immune cells (3); (c) Submucosa: no pathological changes (0), rare (1), moderate (2), or massive invasion of immune cells and edema (3).

**Statistical analyses**. Statistical analysis and graphing were done with Graphpad Prism. Quantitative results are presented as mean values with standard deviation (SD). The two-tailed Student's *t*-test was used to compare two experimental groups. $P < 0.05$ was considered statistically significant.

**Reporting summary**. Further information on research design is available in the Nature Research Reporting Summary linked to this article.

## Data availability
All the sequence data generated for this study have been deposited in NCBI Sequence Read Archive (SRA) under accession number PRJNA558567 and PRJNA557572. The source data underlying Figs. 1c–f, 2b–1, 3a–d, 4a–b and Supplementary Figs. 16–20 are provided as a Source Data file. All data supporting the findings of this study are present in the article and Supplementary Information, or are available from the corresponding author upon reasonable request.

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

## Acknowledgements

This work was supported by National Natural Science Foundation of China (NO. 31700707 and NO. 31870799) and by Natural Science Basic Research Plan in Shaanxi Province of China (NO. 2018JQ2007).

## Author contributions

H.M. and H.B. performed experiments. F.S., Y.L., and C.L. helped to perform animal experiments. Y.Q. and P.C. helped to analyze the high-throughput sequencing results. Y.Y. and L.K. helped to perform data analysis. H.M. and J.D. designed the study, analyzed data, and wrote the manuscript.

## Additional information

**Competing interests:** The authors declare no competing interests.

