## [Peer Review File · Nature Communications]

Reviewers' comments:

Reviewer #1 (Remarks to the Author):

In this study, They introduced a universal strategy for encapsulation of antibiotics by hydrogen sulfide (H₂S) responsive glycovesicles to reduce salmonellosis. However, I do not see that the information provided is not significantly contributing to an advance in technology of cure the salmonellosis and new information of biology. In addition, there is not much of in vivo study to prove the direct evidence of Salmonella inhibition by this construct. It is very difficult to see the direct mechanism to inhibit the target organism specifically by this. It requires a set of precise animal study to show the correlation between the organism and the molecule.

Reviewer #2 (Remarks to the Author):

The manuscript by Mu et al. describes the use of a novel mode to deliver antibiotics (via glycovesicles) with the goal to treat Salmonellosis. The results presented by the authors are novel and promising, however the paper lacks proper descriptions of some aspects of the methodology used. I have additional recommendations that I believe would strengthened the conclusions presented by the authors. My comments are below:

1) It is not clear to me how this mode of delivery would be specifically activated by Salmonella and not by other members of the microbiota that are also able to produce H₂S. The authors show that Listeria doesn't induce the release of the vesicles contents as well as Salmonella, but they should also include another H₂S producing bacteria to their assay. On a similar note, to show specificity to H₂S inducing release, the authors should repeat the experiments with a Salmonella strain that cannot produce H₂S and determine if that reduces the release of the vesicle contents.

2) The authors state that "Pathogen-inspired antimicrobial glycovesicles prevent antibiotic-induced damage to the gut microbiota". However their data shows that treatment with AM-CIP does significantly change the the composition of the gut microbiota (i.e.: Lactobacillus increase) and the authors don't have data to to determine if these changes are beneficial or not to the host. Therefore, the authors statement above is an over interpretation of the data presented and cannot be confirmed unless a functional analysis of the microbiota from all the treatments (i.e.: fecal transplant followed by challenged with Salmonella) is performed.

3) What is the intestinal concentration of the antibiotics (CIP and NEO) when delivered using the vesicles? This information would be very relevant to determine the mode of action of this treatment.

4) Overall the manuscript lacks a proper reference to the statistical methods used to assess differences in the experiments presented. The statistical tests used should be clearly stated on the figure legends and on the material and methods section. On a similar note the microbiota analysis methodology should be expanded to provide detailed information on the softwares and packages used to perform the analysis.

5) Authors should state in the Methodology which Salmonella Typhimurium strain was used in the experiments performed.

6) A quantification of the histopathology analysis of figure 2h should be included as it is hard to see differences in histopathological lesions/inflammation between the groups with the images presented.

7) Figure 1d-f: it is not clear how many times the experiments were performed as how many biological replicates (N)?

Reviewer #3 (Remarks to the Author):

This manuscript describes the glycovesicle system, which releases the antibiotics in response to hydrogen sulfide produced by salmonella. This vesicle system was developed to prevent quick absorption of the antibiotics into bloodstream and allow antibiotics to target H₂S-producing colonies locally at the site of enteric infections. The authors demonstrated that this vesicle system effectively reduced the Salmonella infections in gastrointestinal tract without significantly altering gut microbiota. These results are impressive, but the reviewer feels that the authors should address the following points, which more strongly support the conclusions made by them and warrant publication of this manuscript.

1. The drug releasing experiment shown in Fig. 1d was conducted with 2 mM of hydrogen sulfide. Is this H₂S level biologically relevant and not largely different from the concentration in the H₂S-producing colonies? The authors also should address H₂S concentration-dependent drug releasing ability of AM-CIP.
2. The authors did not provide evidence that the drug concentration increases locally at the infection site and this localized concentration state is maintained for a certain period in the gastrointestinal tract. This is the critical point of the study and thus should be experimentally proved. Fluorescence imaging might be useful to address this point.
3. The fluorophore release experiment should be done with non-infected mice and compare the data with that of the infected mice shown in Figure 2i.
4. The authors claimed that the cleaved xylooligosaccharide diffused in the gastrointestinal tract and promoted the growth of the beneficial microbiota. On the other hand, the authors also claimed that released antibiotics locally remained at the pathogen colonies. The authors should give an explanation about the difference of this molecular behavior.
5. In Fig 3d, AM administration cannot increase the relative abundance of Lactobacillaceae. This result seems to be strange since the vacant vesicle also can be cleaved by hydrogen sulfide and release xylooligosaccharide. The authors should explain this point.
6. In Fig 2b-2f, the drug dosages were not described.

Response to Reviewers' comments:

Reviewer #1 (Remarks to the Author):

In this study, they introduced a universal strategy for encapsulation of antibiotics by hydrogen sulfide (H₂S) responsive glyovesicles to reduce salmonellosis. However, I do not see that the information provided is not significantly contributing to an advance in technology of cure the salmonellosis and new information of biology. In addition, there is not much of in vivo study to prove the direct evidence of Salmonella inhibition by this construct. It is very difficult to see the direct mechanism to inhibit the target organism specifically by this. It requires a set of precise animal study to show the correlation between the organism and the molecule.

Response: Salmonellosis is one of the most frequently occurring intestinal diseases. Although antibiotics are a life-saving tool for the treatment of bacterial infections, antimicrobial therapy is not recommended for this disease. Antibiotic use does not shorten the length of diarrhea, reduces pathogen shedding only transiently, and involves the risk of adverse drug reactions. In this work, we introduce a versatile pathogen-inspired system for delivering absorbable or poorly absorbed antibiotics with a minimal risk of adverse outcomes to treat salmonellosis. This system includes the following advantages: 1) prevent quick absorption of oral antibiotics into the bloodstream; 2) allow antibiotics to target enteric pathogens locally and thereby alleviates disease symptoms; 3) greatly ameliorate antibiotic-induced damage to gut microbiota; and 4) increase beneficial microbiota *Lactobacillaceae* by the releasable prebiotic xylooligosaccharide analogs. This study might open up a new avenue to develop pathogen-inspired antimicrobial nanovesicles to resolve enteric infections.

In this new submission, we performed a set of experiment to address the reviewer's concerns. First of all, we compare the difference in the drug release behavior between *Salmonella* infected and uninfected mice by use of an AM vesicle loaded with Fluo (an H₂S-responsive fluorescence probe). The ileum from infected mice, but not from normal ones elicits the strongest fluorescence intensity in a time-dependent manner (Fig. 2i), implying that 1) *S. typhimurium* triggers the release of the probe from AM vesicles and 2) there is higher relative abundance of *S. typhimurium* in ileum. Secondly, *in vivo* imaging experiments reveal AM vesicles are disassembled on-demand at the site of infection (Fig. 2j). Besides the direct killing of pathogenic bacteria, the increased abundance of *Lactobacillaceae* by the releasable prebiotic xylooligosaccharide analogs might also contribute to the resolution of Salmonellosis. We perform fecal transplant experiments and our data show that gut microbiota shaped by antibiotic-loaded glyovesicle is more beneficial to the host than that by antibiotic alone (Fig. S10).

We hope this reviewer would see our efforts in improving this manuscript and accept this version.

Reviewer #2 (Remarks to the Author):

The manuscript by Mu et al. describes the use of a novel mode to deliver antibiotics (via glyovesicles) with the goal to treat Salmonellosis. The results presented by the authors are novel and promising, however the paper lacks proper descriptions of some aspects of the methodology used. I have additional recommendations that I believe would strengthened the conclusions presented by the authors. My comments are below:

Response: we thank the reviewer for her/his insightful summary and constructive comments. Accordingly, we have made changes upon the comments and substantially improved the manuscript. All changes are tracked, and we hope the reviewer would accept this revised manuscript.

1) It is not clear to me how this mode of delivery would be specifically activated by Salmonella and not by other members of the microbiota that are also able to produce H₂S. The authors show that Listeria doesn't induce the release of the vesicles contents as well as Salmonella, but they should also include another H₂S producing bacteria to their assay. On a similar note, to show specificity to H₂S inducing release, the authors should repeat the experiments with a Salmonella strain that cannot produce H₂S and determine if that reduces the release of the vesicle contents.

Response: as suggested, we include two novel organisms, *S. typhimurium* (producing H₂S) and *C. freundii* (another H₂S-producing organism) in Fig.1f. As shown, There are much stronger fluorescence intensity in the probe-loaded AM vesicles when incubated with both *S. typhimurium* (producing H₂S) than that of *S. paratyphi* A (Fig. 1f), an observation that AM vesicles can only be decomposed by H₂S-producing Salmonella. *S. typhimurium* induces much stronger fluorescence than *C. freundii* does at the same concentration of bacterial cells. The fact that *S. typhimurium* is more abundant at the local sites of infected intestine than other H₂S-producing bacteria will facilitate the disassembly of AM vesicles.

2) The authors state that "Pathogen-inspired antimicrobial glyovesicles prevent antibiotic-induced damage to the gut microbiota". However their data shows that treatment with AM-CIP does significantly change the composition of the gut microbiota (i.e.: Lactobacillus increase) and the authors don't have data to determine if these changes are beneficial or not to the host. Therefore, the authors statement above is an over interpretation of the data presented and cannot be confirmed unless a functional analysis of the microbiota from all the treatments (i.e.: fecal transplant followed by challenged with Salmonella) is performed.

Response: in order not to overinterpret the data, we change the sentence "Pathogen-inspired antimicrobial glyovesicles prevent antibiotic-induced damage to the gut microbiota" as "Pathogen-inspired antimicrobial glyovesicles ameliorate antibiotic-induced damage to the gut microbiota". We performed functional analysis of the microbiota through fecal transplant experiments, as following: To see whether AM-CIP shaped gut microbiota is beneficial to the host, we transfer the gut

microbiota from AM-CIP or CIP treated *Salmonella*-infected donor mice to infected recipient mice. The recipient mice received with AM-CIP shaped gut microbiota have lower pathogen burden and tissue inflammation than that in mice received with CIP shaped one (Fig.S10).

3) What is the intestinal concentration of the antibiotics (CIP and NEO) when delivered using the vesicles? This information would be very relevant to determine the mode of action of this treatment.

Response: the intestinal concentration of CIP is shown in Fig. 3d, and the intestinal concentration of NEO is shown in Fig. S11.

4) Overall the manuscript lacks a proper reference to the statistical methods used to assess differences in the experiments presented. The statistical tests used should be clearly stated on the figure legends and on the material and methods section. On a similar note the microbiota analysis methodology should be expanded to provide detailed information on the softwares and packages used to perform the analysis.

Response: we thank the reviewer's suggestion and we are very sorry for our carelessness. Now we include information about statistical tests in figure legends and method sections. As suggested, the microbiota analysis methodology is also expanded in the method section.

5) Authors should state in the Methodology which *Salmonella Typhimurium* strain was used in the experiments performed.

Response: in this work, three bacteria strains were used, including *Salmonella enterica serovar typhimurium* (SL1344), *Salmonella paratyphi A* (ATCC9150) and *Citrobacter freundii* (ATCC43864). We have added these information in the method section.

6) A quantification of the histopathology analysis of figure 2h should be included as it is hard to see differences in histopathological lesions/inflammation between the groups with the images presented.

Response: now we provide histopathology score of the H&E images in Fig. 2h. The criteria for histopathology scoring is listed in supplementary information.

7) Figure 1d-f: it is not clear how many times the experiments were performed as how many biological replicates (N)?

Response: we have added these information in the legend of Fig. 1.

Reviewer #3 (Remarks to the Author):

This manuscript describes the glycovesicle system, which releases the antibiotics in response to hydrogen sulfide produced by salmonella. This vesicle system was developed to prevent quick absorption of the antibiotics into bloodstream and allow antibiotics to target H₂S-producing colonies locally at the site of enteric infections. The authors demonstrated that this vesicle systems effectively reduced the Salmonella infections in gastrointestinal tract without significant altering gut microbiota. These results are impressive, but the reviewer feel that the authors should address the following points, which more strongly support the conclusions made by them and warrant publication of this manuscript.

Response: we really appreciate the reviewer for her/his insightful summary and constructive comments. Accordingly, we have made changes upon the comments and substantially improved the manuscript. All changes are tracked, and we hope the reviewer would accept this submission.

1. The drug releasing experiment shown in Fig. 1d was conducted with 2 mM of hydrogen sulfide. Is this H₂S level biologically relevant and not largely different from the concentration in the H₂S-producing colonies? The authors also should address H₂S concentration-dependent drug releasing ability of AM-CIP.

Response: The H₂S concentration produced by *S. typhimurium* (10⁶ CFU mL⁻¹) is determined to be 125 μM. We repeat experiments in Fig.1b-1d by use of 100 μM Na₂S. As suggested, we perform the experiment to address H₂S concentration-dependent drug releasing ability of AM-CIP (Fig. S9).

2. The authors did not provide evidence that the drug concentration increase locally at the infection site and this localized concentration state is maintained for certain period in the gastrointestinal tract. This is the critical point of the study and thus should be experimentally proved. The fluorescence imaging might be useful to address this point.

Response: we totally agree with the reviewer's point about this. First, we perform the fluorophore release experiment in non-infected mice (Fig. 2i). The ileum from infected mice, but not from normal ones elicits the strongest fluorescence intensity very quickly (2 h) and the fluorescence increases in a time-dependent manner (Fig. 2i), implying that 1) *S. typhimurium* triggers the release of the probe from AM vesicles and 2) there is higher relative abundance of *S. typhimurium* in ileum. Further, as suggested, we perform fluorescence imaging *in vivo* as following: infected mice are administrated with fluorescence (5-DTAF)-labelled *S. typhimurium* or probe-loaded AM vesicles. The fluorescence-labelled bacteria mainly retain in ileum (Fig. 2j), even after 12 h post oral administration, an observation that the ileum is one primary site of *Salmonella* infection. Consistently, there is strongest fluorescence intensity in ileum 12 h after administration of probe-loaded AM vesicles (Fig. 2j). Taken together, these findings indicate that AM vesicles are disassembled on-demand at the site of infection. We hope the reviewer will accept this improvement.

3. The fluorophore release experiment should be done with non-infected mice and compare the data with that of the infected mice shown in Figure 2i.

Response: we have done this experiment in non-infected mice as shown in Fig. 2i.

4. The authors claimed that the cleaved xylooligosaccharide diffused in gastrointestinal tract and promoted the growth of the beneficial microbiota. On the other hand, the authors also claimed that released antibiotics locally remained at the pathogen colonies. The authors should give explanation about the difference of this molecular behavior.

Response: now we clarify this concern in discussion section (line 208-214, page 11). After cleavage of the disulfide bond, the releasing antibiotic is uptaken by *Salmonella* pathogens nearby and thus dramatically decrease bacterial burden. In contrast, oligosaccharide prebiotics such as xylooligosaccharide could only be utilized by a limited number of beneficial bacteria including lactobacilli and selectively proliferate these organisms.

5. In Fig 3d, AM administration cannot increase the relative abundance of Lactobacillaceae. This result seems to be strange since the vacant vesicle also can be cleaved by hydrogen sulfide and release xylooligosaccharide. The authors should explain this point.

Response: many thanks for the reviewer's carefulness about this. Now explain this point as following:

This is encouraging since several strains from the *Lactobacillaceae* family are shown to be highly antagonistic to *Salmonella* pathogens and protect against *Salmonella* infections in the gastrointestinal tract. It is reasonable that AM treatment doesn't increase *Lactobacillaceae* as much as AM-CIP treatment does, due to the suppressive effect of abundant *Salmonella* pathogens on *Lactobacillaceae* in AM-treated groups. (line 164-168, page 9).

6. In Fig 2b-2f, the drug dosages were not described.

Response: the drug dosages were described in the method section as well as in figure legends of Fig. 2.

REVIEWERS' COMMENTS:

Reviewer #2 (Remarks to the Author):

This is a significantly improved version of the manuscript. The authors have addressed all of my concerns.

Reviewer #3 (Remarks to the Author):

The authors have addressed all of this reviewer's concerns. In particular, the fluorescence imaging (Figure 2i, 2j) newly conducted in the revised manuscript strongly supports the validity of their strategy using glyovesicles in antibiotic therapy. I think the revised manuscript is now suitable for publication.

Response to Reviewers' comments

Reviewer #2 (Remarks to the Author):

This is a significantly improved version of the manuscript. The authors have addressed all of my concerns.

Response: we thank the reviewer's agreement with the improvement of our second submission upon her/his constructive and insightful concerns.

Reviewer #3 (Remarks to the Author):

The authors have addressed all of this reviewer's concerns. In particular, the fluorescence imaging (Figure 2i, 2j) newly conducted in the revised manuscript strongly supports the validity of their strategy using glyovesicles in antibiotic therapy. I think the revised manuscript is now suitable for publication.

Response: we thank the reviewer's agreement with the improvement of our second submission upon her/his constructive and insightful concerns.